# Caregivers’ Emotional Responses Triggered by a False-Positive VLCADD in Newborn Screening in Oita Prefecture

**DOI:** 10.3390/ijns11040090

**Published:** 2025-10-08

**Authors:** Sakura Morishima, Yumi Shimada, Kenji Ihara

**Affiliations:** Department of Pediatrics, Oita University Faculty of Medicine, 1-1 Idaigaoka, Hasama-machi, Yufu City 879-5593, Oita, Japan; sakura0624@oita-u.ac.jp (S.M.); yuminasu@oita-u.ac.jp (Y.S.)

**Keywords:** neonatal screening, VLCADD, false-positive, persistent anxiety

## Abstract

Neonatal screening programs for inborn errors of metabolism are essential for early diagnosis and intervention. However, false-positive results can cause unnecessary psychological stress for caregivers. This study investigated the emotional impact on a small number of caregivers in Oita Prefecture in Japan, whose infants received false-positive screening results for very long-chain acyl-CoA dehydrogenase deficiency (VLCADD). Particular attention was given to caregivers’ concerns regarding episodes of transient fasting suggestive of nutritional deficiency, as well as their perspectives on appropriate feeding practices for newborns. Nineteen infants in Oita Prefecture were identified as having elevated acylcarnitines, which were later confirmed as false positives. Of these cases, 11 mothers consented to participate in a survey and long-term growth evaluation using health check records. Thirty children with normal screening results were included as controls. While no differences in physical growth were found between groups by 3.5 years of age, some mothers of false-positive infants reported persistent anxiety. Their concerns included regret for inadequate breastfeeding and latent adverse effects on long-term growth or development. Conversely, caregivers’ anxiety diminished over time as they directly observed their infants’ normal growth and development. No regret was expressed regarding breastfeeding, and concerns about VLDCAD were not observed. Caregivers’ responses may help reduce their psychological burden.

## 1. Introduction

Neonatal screening programs have been developed for the detection of congenital disorders during a pre-symptomatic stage and are now widely implemented in many countries worldwide. Dried blood spot (DBS) testing has been used for neonatal screenings; and when the results suggest the possibility of a target disorder, caregivers are advised to pursue confirmatory diagnostic evaluations. As a result, false-positive results have been reported to occur with some frequency influenced by contributing factors. For example, it is known that during the diffusion and drying process on filter paper, concentration gradients in spots may occur, potentially reducing measurement accuracy by approximately 10% [1,2,3,4]. To reduce the number of false positives, additional strategies have been explored such as adjusting cut-off values at individual facilities and incorporating genetic testing using next-generation sequencing [5,6,7]. Nonetheless, positive screening results impose psychological distress on some caregivers. Previous studies have shown that parents of infants with positive results from newborn screening had significantly higher Parenting Stress Index (PSI) scores than those of screen-negative infants, with elevated stress persisting for up to one year [8]. Their rate of pediatric clinic visits was also significantly higher. It has been reported that receiving the possible diagnosis of any disease in infants may heighten parental anxiety, strain family relationships, and amplify concerns about the child’s health, even when pediatricians later explained the false positive results of metabolic disorders [9]. Notably, the levels of parental understanding regarding the reason for re-testing were inversely correlated with stress levels—the lower the understanding, the greater the stress [10]. Multiple factors have also been reported as contributing to caregivers’ anxiety. These include the prolonged waiting period for follow-up or confirmatory test results, which extends emotional distress; characteristics of the disorder, such as its severity and the availability of treatment; inadequate explanations from healthcare providers regarding false-positive results and variations in caregiver comprehension; and differences in social infrastructure, including healthcare systems [9,11,12].

The frequency and causes of false positives vary depending on the target diseases [13,14]. Regarding very long-chain acyl-CoA dehydrogenase deficiency (VLCADD), elevated C14 and C14:1/C2 ratios have been observed in infants with significant postnatal weight loss at the time of blood collection [2]. In our previous research, environmental factors contributing to excessive weight loss include exclusive breastfeeding and early rooming-in practices [15]. Fasting induced by breastfeeding may contribute to false-positive results for VLCADD. Moreover, babies with false-positive VLCADD results associated with early excessive weight loss might experience long-term impacts on their growth or development. Consequently, caregivers may experience regret over having unintentionally placed their newborn in a fasting state, which may have led to changes in neonatal nutrition, particularly in breastfeeding. The uncertainty regarding long-term developmental outcomes may further exacerbate their anxiety.

This study aimed to examine the emotional changes experienced by a small number of caregivers of infants in Oita Prefecture in Japan who received false-positive results for VLCADD. Additionally, it sought to assess how observations of the child’s subsequent growth and development influenced caregivers’ perspectives on neonatal nutrition, particularly exclusive breastfeeding, and whether these perspectives evolved over time. For this purpose, nutritional status was first assessed retrospectively using a structured questionnaire. Subsequently, face-to-face interviews were conducted to explore whether caregivers experienced nutrition-related anxiety, the specific aspects of nutrition that contributed to such anxiety, and how these concerns changed over time.

## 2. Materials and Methods

### 2.1. Ethical Considerations

This study was approved by the Ethics Committee of the Faculty of Medicine, Oita University (Approval No. 2542 on 23 May 2023) and Almeida Hospital (Approval No. 221 on 5 June 2023).

### 2.2. Study Population

#### 2.2.1. Extraction of Archived Data via Tandem Mass Spectrometry

Under ethical approval for this study, we reviewed archived newborn screening data stored at the local screening facility at Almeida Hospital/Laboratory Center. We identified neonates born in Oita Prefecture between April 2014 and March 2024 whose VLCADD screening data exceeded cut-off values (C14:1 > 0.25 μmol/L and C14:1/C2 ratio > 0.013) but were subsequently determined normal following re-testing and confirmatory examinations. These cases were classified as false positives.

#### 2.2.2. Enrollment of Parents of the False-Positive Infants for Interviews

Parents of the false-positive infants were contacted by mail and invited to participate in the study. The parents who provided written informed consent to us were interviewed either in person or by mail. During face-to-face interviews, the content of the completed questionnaires was reviewed to assess caregivers’ anxiety regarding the transient fasting state from nutritional deficiency, as well as to examine any subsequent changes in their perspectives on related feeding practices.

For the families who could not be reached by our study proposal, an opt-out notice was posted on the institutional website, and only minimal data extracted from medical records in the Almeida hospital/laboratory center were used for analysis. In addition, neonatal screening data of tandem mass spectrometry from 30 healthy infants who had provided written consent to participate in the study were assigned as negative controls.

#### 2.2.3. Enrollment of Healthy Control Participants for Interviews

The parents of children who had received routine vaccinations and completed their 3.5-year health checkup at a pediatric clinic (Oita Children’s Hospital) in Oita City were invited to participate in the study. After obtaining written informed consent, we reviewed the maternal and child health handbooks and conducted interviews focusing primarily on breastfeeding practices. For participants who provided informed consent, tandem mass spectrometry data stored at the neonatal screening facility were retrieved. Acylcarnitine and amino acid profiles were then used to compare the false positive and the control groups.

#### 2.2.4. Growth Record Collection

Data on body weight at birth and at the time of neonatal screening, feeding methods during infancy, and anthropometric measurements at 1 month, 3–4 months, 9–11 months, 18 months, and 3.5 years of age were collected from the maternal and child health handbooks.

#### 2.2.5. Questionnaire Survey

The questionnaire items were structured as follows:

**For all participants** (multiple-choice):Mothers’ attitudes toward breastfeeding during pregnancy;Feeding method at the time of newborn screening;Breastfeeding and infant care practices at the obstetric facility.

**For caregivers of false-positive infants only** (multiple-choice [multiple answers allowed]):Emotional responses after learning that the screening result was a false positive;Emotional reactions upon being informed that confirmatory testing was needed;Reasons for choosing the maternity hospital (multiple-choice);Mothers’ retrospective views on breastfeeding in the maternity hospital;Evaluation of breastfeeding instruction received at the maternity hospital.

### 2.3. Statistical Analysis

All statistical analyses were performed using SPSS 24 (IBM, Tokyo, Japan). The Shapiro–Wilk test was used to assess data normality. For normally distributed variables, group comparisons were conducted using the Student’s *t*-test. For non-normally distributed data, the Mann–Whitney U test was used.

## 3. Results

### 3.1. Participant Analysis

Among the 19 false positives, the caregivers of 11 infants consented to participate in the questionnaire-based survey. Only data from tandem mass spectrometry were analyzed for the remaining eight infants. Among the caregivers of 11 infants, six participated in both the questionnaire survey and individual interviews based on their responses, while five completed only the questionnaire survey via mail.

The mean birth weight in the false-positive and screen-negative groups was 3061.3 g and 3052.0 g, respectively, with no statistically significant difference and both values being consistent with national averages for Japanese newborns. However, at the time of newborn screening blood collection, the mean body weight was significantly lower in the false-positive group (2811.2 g) compared to the negative group (2946.0 g). The rate of weight loss was significantly greater in the false-positive group (−9.2%) than in the negative group (−3.9%) (Table 1).

Despite these early differences, no statistically significant differences were observed between the groups in growth outcomes assessed at 1 month, 3–4 months, 9–11 months, 18 months, and 3.5 years of age (Table 2). In addition, no notable abnormalities in growth or development were identified in any of the six participants from the false-positive group who underwent interviews or the 30 participants in the control group, as confirmed through maternal and child health handbooks.

### 3.2. Questionnaire Survey

Participants were asked about their attitudes toward breastfeeding during pregnancy (Table 3). Both groups showed similar tendencies, with approximately 90% of respondents in each group expressing a desire to breastfeed as much as possible. Nonetheless, at the time of neonatal screening, the proportion of exclusively breastfed infants was higher in the false-positive group (6 of 11; 55%) than in the negative group (4 of 30; 13%).

Furthermore, rooming-in with the mother immediately after birth was more common in the false-positive group (6 of 11; 55%) compared to the screen-negative group (2 of 30; 7%), suggesting a possible association with higher exclusive breastfeeding rates. No significant differences were observed between the two groups in feeding methods at 1 and 3 months of age, nor in the proportion of participants who received breastfeeding guidance at the birth hospital.

Regarding parental emotional responses to the false-positive results, 3 of the 11 (27%) caregivers in the false-positive group reported continued anxiety even after being informed that their child did not have a metabolic disorder, indicating that false-positive results may contribute to lingering psychological distress. Among the caregivers who participated in interviews, some expressed concerns such as, “Each time I visited the pediatrician, I wondered whether it was related to the false-positive result,” and, “I was worried that the significant weight loss might impact future development.” However, only 1 of the 11 caregivers (9%) expressed regret regarding breastfeeding as a method of nutrition. There were also statements indicating that vague anxiety gradually diminished as normal growth was observed.

There were no significant differences between the groups in terms of feeding methods at 1 and 3 months of age or in the level of breastfeeding support received at the birth facility. However, a higher proportion of mothers in the false-positive group (82%) reported difficulties with breastfeeding compared to those in the screen-negative group (53%).

When asked why they chose their delivery facility, 45% of mothers in the false-positive group cited the facility’s active promotion of breastfeeding, which may be related to the higher rate of exclusive breastfeeding during hospitalization. Additionally, 72% of mothers in the false-positive group expressed retrospective concern that formula supplementation might not have been appropriate, compared to 40% in the screen-negative group, suggesting underlying concerns about insufficient milk supply.

## 4. Discussion

The questionnaire responses by the caregivers following mini-interviews revealed lingering psychological distress even after families were informed that the screening test was a false positive, i.e., normal results. Specifically, some parents expressed concerns that their child might later require any pediatric care, worrying that the underlying health issue might be related to the initial “re-test requirement” by the neonatal screening—even though it had been proven to be a false positive. However, no regret was expressed regarding breastfeeding itself, and no specific anxiety related to VLCADD was identified. Although a sense of vague anxiety had been present, it appeared to gradually diminish and fade from memory as the child experienced healthy growth and development.

The infants in the false-positive group exhibited significantly greater postnatal weight loss at the time of neonatal screening blood collection, whereas no significant differences in physical growth were observed at the 1-month checkup, suggesting the presence of adequate catch-up growth. These findings indicate that false-positive infants we studied subsequently demonstrated standard growth trajectories in both height and weight, implying that transient weight loss did not adversely impact later physical development. The process of confirming healthy development over time was considered to have contributed to the alleviation of anxiety.

These findings suggest that consultant pediatricians or neonatologists should consider offering supplemental information to families prior to confirmatory testing, along with explaining the severity and treatability of VLCADD. This could include an explanation that significant postnatal weight loss—especially in exclusively breastfed infants—can influence acylcarnitine levels, resulting in a false-positive result for VLCADD, and is not indicative of any underlying disorders. Emphasizing that false-positive infants do not experience long-term problems with physical growth may further help alleviate caregiver anxiety and provide encouragement.

Some mothers in the false-positive group expressed retrospective regret that they had not supplemented with formula, particularly in the context of significant early weight loss. While breastfeeding is known to confer immunological benefits and important initial attachment between mothers and their children, it is also associated with an increased risk of rehospitalization for dehydration or jaundice in some settings [16,17]. Therefore, personal feeding guidance from a multidisciplinary team during the neonatal period would ensure optimal nutrition and minimize unnecessary parental distress.

## 5. Limitations

This study has several limitations. First, the sample size was small, and responses were collected at a single time point. To address this point, it is necessary to either extend the study period to increase the number of cases at our institution or conduct a multicenter collaborative analysis to accumulate a larger sample, particularly to reassess the potential association between neonatal weight loss and false-positive VLCADD results. Second, as participants were asked to recall past experiences, the findings may have been affected by recall bias. Moreover, the wide age range of participants at the time of the survey means that those surveyed long after the neonatal screening may have had blurred memories of the emotions and anxieties experienced during that period. In addition, changes in emotional responses over time could not be assessed since this was not a longitudinal study. Finally, the home-made questionnaire was developed only for this study and was administered solely by physicians. In light of these limitations, future research should involve perspectives from other healthcare professionals, such as midwives, to gain a more comprehensive understanding of family experience following false-positive newborn screening results. Validated instruments such as the Parental Stress Index may serve as appropriate tools to generate reliable evidence to inform future policy and practice.

## 6. Conclusions

Despite being informed of the false-positive result, some caregivers continued to experience residual anxiety. Even after confirmation of the false-positive result, caregivers experienced persistent concerns regarding nutritional status and growth, as well as non-specific anxiety. However, these concerns were alleviated over time as they observed their child’s normal growth and development. No regret regarding breastfeeding was expressed, and disease-specific anxiety—such as fear related to the risk of sudden death associated with VLCADD—was not identified among the participants.

These findings should be shared with obstetricians and midwives involved in newborn screening to promote collaborative discussion on appropriate nutritional guidance and strategies to alleviate parental anxiety. Continued interdisciplinary efforts are warranted to optimize both infant health outcomes and family psychological well-being.

## Figures and Tables

**Table 1 IJNS-11-00090-t001:** Comparison up to newborn screening tests.

Variable		False Positive	Screen Negative
Caregivers (*n*)		19	30
Male–Female		12:7	18:12
Gestational age (median)		39w3d	39w6d
At birth	Weight		
	mean, g (SD)	3015.6 (±220.9)	3136.6 (±358.5)
	median, g (IQR)	3078.0 (2912–3215)	3145 (2875–3235)
	Length		
	mean, cm (SD)	48.8 (±2.0)	49.5 (±1.7)
	median, cm (IQR)	49.3 (48.2–50.2)	48.8 (48.5–50.5)
At the time of blood sampling	Day of life	5	5
	Body weight		
	mean, g (SD)	2758.22 (±246.7)	3078.4 (±352.1)
	median, g (IQR)	2868 (2657–2960)	3022 (2788–3360)
	Weight loss, %	−9.2	−2.7
	Breastfeeding, *n* (%)	6 (55%)	4 (13%)
	Mixed feeding, *n* (%)	5 (45%)	26 (87%)
	Formula only, *n*	0	0

SD: Shows how far away from the average.

**Table 2 IJNS-11-00090-t002:** Comparison of the physical growth between the false-negative and screen-negative groups up to the age of 3 years.

		False Positive	Screen Negative	*p*-Value
1 month	Body weight, g (SD)	3842.9 (−0.49)	4143.7 (−0.024)	0.06
	Weight gain, g/day	44.2	40.9	0.40
	Height, cm (SD)	52.7 (+0.064)	53.1 (+0.11)	0.45
3–4 months	Body weight, g(SD)	6565.0 (−0.12)	6279.0 (−0.11)	0.27
	Height, cm (SD)	62.5 (−0.25)	61.1 (−0.24)	0.067
9–11 months	Body weight, g (SD)	8515 (−0.26)	8527.4 (−0.38)	0.59
	Height, cm (SD)	71.6 (−0.34)	70.9 (−0.74)	0.28
18 months	Body weight, g (SD)	9.8 (−0.29)	10.3 (+0.013)	0.25
	Height, cm (SD)	77.9 (−0.60)	79.1 (−0.55)	0.39
3 years	Body weight, g (SD)	13.8 (−0.036)	14.0 (−0.037)	0.67
	Height, cm (SD)	94.5 (−0.38)	94.7 (−0.35)	0.51

**Table 3 IJNS-11-00090-t003:** Comparison of the questionnaire between the false-negative and screen-negative groups.

Question		False Positive	Screen-Negative
**(1) Attitudes Toward Breastfeeding** (Multiple choices allowed)			
“*I definitely wanted to breastfeed*”, Yes (%)		5 (45%)	11 (37%)
“*I wanted to breastfeed if I could produce enough milk*”, Yes (%)		5 (45%)	16 (53%)
“*I preferred to formula-feed*”, Yes (%)		0 (0%)	0 (0%)
“*I had no particular thoughts*”, Yes (%)		1 (9%)	3 (10%)
**(2) Support Practice**			
		Yes	Yes
Breastfeeding initiated within 30 min after birth, Yes (%)		6 (55%)	10 (33%)
Rooming-in with the mother immediately after delivery, Yes (%)		6 (55%)	2 (7%)
Breastfed the baby on demand (whenever the baby wanted to feed), Yes (%)		10 (91%)	20 (67%)
**(3) Feeding methods at 1 and 3 months of age (breastfeeding/mixed feeding/formula only)**			
At 1 month	Breastfeeding, Yes (%)	5 (45%)	14 (47%)
	Mixed feeding, Yes (%)	6 (55%)	16 (53%)
	Formula only, Yes (%)	0 (0%)	0 (0%)
At 3 months	Breastfeeding, Yes (%)	5 (45%)	17 (57%)
	Mixed feeding, Yes (%)	5 (45%)	7 (23%)
	Formula only, Yes (%)	1 (9%)	6 (20%)
**(4) Breastfeeding instruction received from healthcare providers**			
During pregnancy	Received, Yes (%)	10 (91%)	23 (77%)
	Not received, Yes (%)	1 (9%)	6 (20%)
	No opportunity, Yes (%)	0 (0%)	1 (3%)
Postpartum	Received, Yes (%)	10 (91%)	28 (93%)
	Not received, Yes (%)	1 (9%)	1 (3%)
	No opportunity, Yes (%)	0 (0%)	1 (3%)
**(5) Experience of breastfeeding difficulties**			
Throughout infancy	Yes, (%)	9 (82%)	16 (53%)
**(6) Reasons for choosing the delivery facility** **(Multiple choices allowed)**			
Proximity to home (or family home), Yes (%)		7 (64%)	16 (53%)
Good reputation, Yes (%)		2 (18%)	7 (23%)
Clean and sanitary facility, Yes (%)		4 (36%)	7 (23%)
Well-equipped services, Yes (%)		2 (18%)	8 (27%)
Support for breastfeeding, Yes (%)		5 (45%)	0
Other reasons, (%)		4 (36%)	14 (47%)
**(7) Reflections on breastfeeding experience at the delivery facility** **(Multiple choices allowed)**			
“*I’m glad I was able to breastfeed from the newborn period*”, Yes (%)		7 (64%)	20 (67%)
“*It was difficult because I was unable to produce sufficient milk*”, Yes (%)		5 (45%)	11 (37%)
“*I appreciated the adequate guidance on breastfeeding*”, Yes (%)		3 (27%)	8 (27%)
“*I wish I had supplemented with formula when I couldn’t produce sufficient breast milk*”, Yes (%)		8 (72%)	12 (40%)
Others, (%)		2 (18%)	2 (7%)
**(8) Feelings upon receiving a notice for further testing** **for false positives (multiple choices allowed)**		False positive (*n* = 11)	
“*I was shocked*”, Yes (%)		6 (55%)	
“*I was afraid that my baby might have a disease*”, Yes (%)		9 (82%)	
“*I thought it might have been an error*”, Yes (%)		2 (18%)	
“*I will consult a doctor because the result seemed inconclusive*”, Yes (%)		3 (27%)	
“*I did not comprehend its meaning*”, Yes (%)		0 (0%)	
Others, (%)		1 (9%)	
**(9) The emotional response upon learning that the result was a false positive caused by starvation (multiple choices allowed)**			
“*I felt relieved*”, Yes (%)		9 (82%)	
“*I remained concerned whether everything was indeed fine*”, Yes (%)		3 (27%)	
“*I regretted not giving formula when the baby was hungry*”, Yes (%)		1 (9%)	
Other, (%)		2 (18%)	
**(10) Emotional reactions upon receiving normal newborn screening results for screen-negatives (Multiple Answers Allowed)**		Screen negative (*n* = 30)	
“*I was assured that the result was normal*”, Yes (%)		26 (87%)	
“*I had forgotten that the test had been conducted*”, Yes (%)		4 (13%)	
Others, (%)		0 (0%)	

## Data Availability

The data analyzed in this study are not publicly available due to concerns regarding participant privacy. However, all data supporting the findings of this study are available from the corresponding author (k-ihara@oita-u.ac.jp) upon reasonable request.

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
