# Peer review of "Caregivers’ Emotional Responses Triggered by a False-Positive VLCADD in Newborn Screening in Oita Prefecture"

_2409-515X, 2025, doi:10.3390/ijns11040090_

Round 1

Reviewer 1 Report

Comments and Suggestions for Authors

  1. Since the title is 'ParentaI Anxiety caused by false-positive results in newborn screening', one would expect that there would be review of articles on similar issues that have been published. On the contrary, the references are heavy on the technical aspects of newborn screening and false positives on newborn screening.

There are several papers that focus on psychosocial factors on false positives on newborn screening. Here are some. Cross references will show more.

Beucher J., Leray E., Deneuville E., Roblin M., Pin I., Bremont F., Turck D., Giniès J.-L., Foucaud P., Rault G., et al. Psychological Effects of False-Positive Results in Cystic Fibrosis Newborn Screening: A Two-Year Follow-Up. J. Pediatr. 2010;156:771–776.e1. doi: 10.1016/j.jpeds.2009.12.003. [DOI] [PubMed] [Google Scholar]

Tluczek A, Ersig AL, Lee S. Psychosocial Issues Related to Newborn Screening: A Systematic Review and Synthesis. Int J Neonatal Screen. 2022 Sep 27;8(4):53. doi: 10.3390/ijns8040053. PMID: 36278623; PMCID: PMC9589938

Tu WJ, He J, Chen H, Shi XD, Li Y. Psychological effects of false-positive results in expanded newborn screening in China. PLoS One. 2012;7(4):e36235. doi: 10.1371/journal.pone.0036235. Epub 2012 Apr 27. PMID: 22558398; PMCID: PMC3338668.

Hewlett J, Waisbren SE. A review of the psychosocial effects of false-positive results on parents and current communication practices in newborn screening. J Inherit Metab Dis. 2006 Oct;29(5):677-82. doi: 10.1007/s10545-006-0381-1. Epub 2006 Aug 17. PMID: 16917730.

Bodegård G, Fyrö K, Larsson A. Psychological reactions in 102 families with a newborn who has a falsely positive screening test for congenital hypothyroidism. Acta Paediatr Scand Suppl. 1983;304:1-21. doi: 10.1111/j.1651-2227.1983.tb09850.x. PMID: 6576613.

  1. This paper uses the same data in the paper entitled ‘Characteristic Findings of Infants with Transient Elevation of Acylcarnitines in Neonatal Screening and Neonatal Weight Loss. Int. J. Neonatal Screen. 2025,11,33. with the same authors (Morishima, S.; Shimada, Y.; Watanabe, Y.; Ihara, K. There was an extensive discussion and this paper stated that ‘This study had several limitations. Perinatal maternal factors and fetal growth rates were not examined. Additionally, feeding conditions at the maternity hospital were as- sessed based on maternal interviews and maternal health record entries, suggesting that the actual feeding volume, feeding intervals, and maximum weight loss were not objectively measured’.

If the author wishes to pursue the angle of parental anxiety, metrics on anxiety should be focus of this paper. You can expand the part on Parenting Stree Index.

  1. Line 32 – ‘inborn screenings’ is not the usual term. The recommended term is ‘newborn screening’. The next line is erroneous. Newborn screening is NOT diagnostic. As a rule, there is a recommendation of a confirmatory testing.
  2.  
  3. This paper was on parental anxiety on false positive on NBS but the pape rstated that the interviews focused on breastfeeding practices. Line 87-89 ‘After obtaining written informed consent, we reviewed the maternal and child health handbooks and conducted interviews focusing primarily on breastfeeding practices.’

The questionnaire  was heavy on questions about breastfeeding but there were 2 questions referring to emotional responses to false positive test.  Since the scope of the study was 10 years, how did authors manage the accuracy of these responses from the early years of the study.

  1. Results do not show substantial basis for parental anxiety as a cause of false positive results.
  2. Conclusion cannot adequately answer the objective of the study…. This study aimed to investigate the underlying effects of false-positive results of VLCADD screening on subsequent growth in infants.

Author Response

(Comment1) Since the title is 'ParentaI Anxiety caused by false-positive results in newborn screening', one would expect that there would be review of articles on similar issues that have been published. On the contrary, the references are heavy on the technical aspects of newborn screening and false positives on newborn screening. There are several papers that focus on psychosocial factors on false positives on newborn screening. Here are some. Cross references will show more.

    1. Beucher J., Leray E., Deneuville E., Roblin M., Pin I., Bremont F., Turck D., Giniès J.-L., Foucaud P., Rault G., et al. Psychological Effects of False-Positive Results in Cystic Fibrosis Newborn Screening: A Two-Year Follow-Up. J. Pediatr. 2010;156:771–776.e1. doi: 10.1016/j.jpeds.2009.12.003. [DOI] [PubMed] [Google Scholar]
    2. A, Ersig AL, Lee S. Psychosocial Issues Related to Newborn Screening: A Systematic Review and Synthesis. Int J Neonatal Screen. 2022 Sep 27;8(4):53. doi: 10.3390/ijns8040053. PMID: 36278623; PMCID: PMC9589938
    3. Tu WJ, He J, Chen H, Shi XD, Li Y. Psychological effects of false-positive results in expanded newborn screening in China. PLoS One. 2012;7(4):e36235. doi: 10.1371/journal.pone.0036235. Epub 2012 Apr 27. PMID: 22558398; PMCID: PMC3338668.
    4. Hewlett J, Waisbren SE. A review of the psychosocial effects of false-positive results on parents and current communication practices in newborn screening. J Inherit Metab Dis. 2006 Oct;29(5):677-82. doi: 10.1007/s10545-006-0381-1. Epub 2006 Aug 17. PMID: 16917730.
    5. Bodegård G, Fyrö K, Larsson A. Psychological reactions in 102 families with a newborn who has a falsely positive screening test for congenital hypothyroidism. Acta Paediatr Scand Suppl. 1983;304:1-21. doi: 10.1111/j.1651-2227.1983.tb09850.x. PMID: 6576613.

(Answer) In the previously published studies you kindly shared, caregivers expressed anxiety related to the prolonged waiting period for test results, the severity of disease even in cases of false negatives, and vague concerns about long-term prognosis influenced by their level of understanding.

In contrast, the false-positive cases of VLCADD examined in this study were attributed to elevated acylcarnitine levels resulting from weight loss due to insufficient nutrition. Based on this, we hypothesized that caregivers might experience feelings of self-blame regarding feeding practices—particularly exclusive breastfeeding—and anxiety about the potential effects on future growth and development. This content has been included in lines 68–71 of the Introduction.

(Comment 2) This paper uses the same data in the paper entitled ‘Characteristic Findings of Infants with Transient Elevation of Acylcarnitines in Neonatal Screening and Neonatal Weight Loss. Int. J. Neonatal Screen. 2025,11,33. with the same authors (Morishima, S.; Shimada, Y.; Watanabe, Y.; Ihara, K. There was an extensive discussion and this paper stated that ‘This study had several limitations. Perinatal maternal factors and fetal growth rates were not examined. Additionally, feeding conditions at the maternity hospital were as- sessed based on maternal interviews and maternal health record entries, suggesting that the actual feeding volume, feeding intervals, and maximum weight loss were not objectively measured’.

If the author wishes to pursue the angle of parental anxiety, metrics on anxiety should be focus of this paper. You can expand the part on Parenting Stree Index.

(Answer) The PSI, as you pointed out, was not evaluated in this study. Since the answers in the questionnaire were based on retrospective recall from the caregivers, there existed potential inaccuracy due to recall bias, as you have indicated. To help mitigate this limitation, we conducted direct interviews to explore changes in maternal emotional responses through interactive, conversational questioning. This approach has been described in the Methods section (line 97-100).

(Comment 3) Line 32 – ‘inborn screenings’ is not the usual term. The recommended term is ‘newborn screening’. The next line is erroneous. Newborn screening is NOT diagnostic. As a rule, there is a recommendation of a confirmatory testing.

(Answer) As you pointed out, we have revised the term to “newborn screening test.” We also acknowledge your point that screening is not equivalent to diagnosis. Accordingly, we have modified the expression as follows: “and when the results suggest the possibility of a target disorder, caregivers are advised to pursue confirmatory diagnostic evaluations.” This revision has been made in lines 36–38 of the Introduction.

(Comment 4) This paper was on parental anxiety on false positive on NBS but the pape rstated that the interviews focused on breastfeeding practices. Line 87-89 ‘After obtaining written informed consent, we reviewed the maternal and child health handbooks and conducted interviews focusing primarily on breastfeeding practices.’

The questionnaire  was heavy on questions about breastfeeding but there were 2 questions referring to emotional responses to false positive test.  Since the scope of the study was 10 years, how did authors manage the accuracy of these responses from the early years of the study.

(Answer) As you pointed out, the questionnaire focused primarily on neonatal feeding practices, particularly breastfeeding. This focus was based on previous reports indicating that infants with significant weight loss are more likely to receive false-positive VLCADD results, and that exclusive breastfeeding may contribute to greater weight loss in cases of insufficient milk intake. Based on this, we hypothesized that the proportion of exclusively breastfed infants would be higher among those with false-positive results and therefore designed the questionnaire to emphasize breastfeeding-related items.

As you also pointed out, the duration between the neonatal period and the time of interview varied substantially across cases (ranging from 9 months to 9 years and 6 months). For caregivers who surveyed after a longer interval, then memories might have faded and emotional responses had changed over time. We recognize this recall bias as a limitation of the present study and have addressed it accordingly (line 236-238).

(Comment 5) Results do not show substantial basis for parental anxiety as a cause of false positive results.

(Answer) As you pointed out, this study did not objectively assess anxiety using specific materials such as the PSI. Instead, we used a questionnaire-based interview to explore the specific sources of caregiver anxiety. The findings indicated that the anxiety was not attributable to the nature of the disease itself, such as the risk of sudden death. Rather, caregivers reported concerns related to the child’s growth and nutritional status, as well as, vague or non-specific anxiety. These concerns were found to gradually diminish over time as caregivers observed normal developmental progress in their children.

(Comment 6) Conclusion cannot adequately answer the objective of the study…. This study aimed to investigate the underlying effects of false-positive results of VLCADD screening on subsequent growth in infants.

(Answer) As you rightly noted, the study objective has been revised to: “To clarify the emotional changes in mothers who were informed that the cause of a false-positive VLCADD result was presumed to be fasting.” The study title has also been changed to: “Characteristics of caregivers’ emotional response triggered by a false-positive VLCADD in neonatal screening.”

The growth and development of infants with false-positive VLCADD results were confirmed to have been within normal ranges by review of maternal and child health handbooks. Moreover, caregiver anxiety seemed to diminish over time, particularly as they observed their child’s normal growth and developmental progress.

Although being informed of substantial weight loss due to fasting—causing the false-positive results—may have brought parental anxiety, our findings indicate that such transient fasting and weight loss did not negatively affect subsequent development. We believe that sharing these findings with caregivers of the similar cases in future may help to alleviate their anxiety. These points will be incorporated into the Conclusion sections (line 255-257).

Reviewer 2 Report

Comments and Suggestions for Authors

Dear Editor,

Thank you for the opportunity to review the manuscript titled "Parental anxiety caused by false-positive results in newborn screening." I appreciate the authors' efforts in addressing an important and often overlooked psychosocial aspect of newborn screening programs. Below are my detailed comments and suggestions to improve the clarity, coherence, and scientific rigor of the manuscript.

  1. Study Objective (Line 56):
    The stated primary objective—“to investigate the underlying effects of false-positive results of VLCADD screening on subsequent growth in infants”—does not appear to align with the manuscript’s title, which focuses on parental anxiety. The authors should clarify whether the main outcome is psychological (parental anxiety) or physical (infant growth). If both are addressed, the rationale and prioritization of these outcomes should be clearly stated. Please revise either the objective or the title accordingly for consistency.
  2. Rationale for the Study (Line 56):
    The manuscript would benefit from a clearer justification for the study. What is the underlying rationale, supported by existing literature or preliminary data, that motivated this investigation? Specifically, how common and impactful is parental anxiety caused by false-positive VLCADD screening results, and what are the potential long-term consequences? Please provide relevant references or background information to strengthen the rationale.
  3. Recruitment of Healthy Controls (Line 84):
    The process for enrolling healthy control participants needs clarification. Did the authors review newborn screening results to ensure that control infants had definitively negative results for VLCADD and other screened conditions? If so, this should be clearly stated in the Methods section to support the validity of group comparisons.
  4. Use of Questionnaire and Recall Bias (Line 95):
    Given that this is a retrospective study involving caregiver-reported emotional responses, recall bias is a significant concern. The manuscript should address this limitation explicitly, as parental recollection of emotions during the newborn period may be inaccurate or diminished over time.
  5. Emotional Impact Timing and Measurement:
    The assessment of emotional responses at a single time point may not fully capture the dynamic nature of psychological reactions. Emotions such as anxiety or regret often fluctuate and may resolve or intensify over time. The authors should acknowledge that evaluating emotional outcomes longitudinally would provide a more comprehensive understanding of psychological burden. This limitation should be discussed.
  6. Questionnaire Content:
    It is unclear whether the questionnaire included items assessing long-term concerns of the parents regarding their child’s growth, development, or health outcomes following the false-positive result. If not, this may represent an important missed opportunity, as ongoing worry is a key component of parental anxiety. Please clarify the scope of the emotional domains evaluated.
  7. Emotional Recall Validity:
    In assessing emotional reactions such as anxiety or confusion, did the authors consider the possibility that caregivers may have forgotten or minimized their feelings at the time of the event? This issue further underscores the need to address recall bias and emotional adaptation over time.
  8. Breastfeeding Status and VLCADD False Positives:
    The rationale for including breastfeeding status in the analysis of VLCADD false positives should be clarified. Is there evidence suggesting that breastfeeding may influence acylcarnitine profiles or screening accuracy? If so, please provide supporting references and explain how this variable was incorporated into the analysis.
  9. Table 1 – Descriptive Statistics:
    Given the relatively small sample sizes, it would be helpful to report both mean ± SD and median and IQR for continuous variables, especially those with skewed distributions. This will provide a more complete picture of central tendency and variability.
  10. Table 1 – Standard Deviation Format:
    The SD should be reported using the conventional “±” format. For example, the value “3061.3 (+0.052)” is unclear: does “+0.052” represent the SD, or a deviation from a reference value? If this is the SD, please revise it to “3061.3 ± 0.052.” If it refers to a difference from a normal/reference value, this should be clarified in the table or accompanying legend to avoid misinterpretation.
  11. Table 3 – Formatting and Interpretation:
  • The label “Table 3” should be bold.
  • Additionally, the discussion and interpretation of the questionnaire results (questions 1–7) presented in Table 3 are insufficient.
  1. This study is likely central to understanding the parental emotional experience, yet they are underexplored in the results and discussion. The authors should expand on the meaning, implications, and possible psychological interpretations of these items in the context of false-positive screening results.
  2. Reference Formatting:
    Please ensure that all references are formatted in accordance with the journal’s guidelines.

Summary

This study addresses a valuable issue related to newborn screening; however, revisions are needed to improve the alignment between objectives and outcomes, strengthen the methodological transparency, and enhance the clarity of data presentation. Addressing the above points will significantly improve the manuscript's quality and impact.

Author Response

(Comment 1) Study Objective (Line 56):
The stated primary objective—“to investigate the underlying effects of false-positive results of VLCADD screening on subsequent growth in infants”—does not appear to align with the manuscript’s title, which focuses on parental anxiety. The authors should clarify whether the main outcome is psychological (parental anxiety) or physical (infant growth). If both are addressed, the rationale and prioritization of these outcomes should be clearly stated. Please revise either the objective or the title accordingly for consistency.

(Answer) Thank you for your insightful comment. As you pointed out, addressing two distinct themes may lead to confusion for readers. In response, we revised the title to focus primarily on caregiver anxiety, placing particular emphasis on emotional responses related to feeding practices, especially breastfeeding. The revised title is: “Characteristics of caregivers’ emotional response triggered by a false-positive VLCADD in newborn screening”

(Comment 2)  Rationale for the Study (Line 56):
The manuscript would benefit from a clearer justification for the study. What is the underlying rationale, supported by existing literature or preliminary data, that motivated this investigation? Specifically, how common and impactful is parental anxiety caused by false-positive VLCADD screening results, and what are the potential long-term consequences? Please provide relevant references or background information to strengthen the rationale.

(Answer) Thank you for your comment. As our research hypothesis, we assumed that insufficient milk supply for infants under exclusive breastfeeding, the early postpartum period may have been at transient fasting state, resulting in elevated levels of specific acylcarnitines. This could potentially cause some items at exceed the screening cutoff for VLCADD in tandem mass spectrometry. Based on this assumption, we examined the relationships among these factors.

Previous studies have reported that infants with false-positive VLCADD results tend to exhibit greater weight loss at the time of screening. Separately, several studies have addressed caregivers’ anxiety following false-positive screening results with the advancement of technologies such as expanded mass screening and the increasing scope of pre-symptomatic neonatal diagnoses. In the present study, we hypothesized that false-positive VLCADD results may give rise to disease-specific anxiety, particularly concerning breastfeeding, and aimed to explore how such anxiety changes over time. This content has been added to lines 71–80 of the Introduction.

(Comment 3) Recruitment of Healthy Controls (Line 84):
The process for enrolling healthy control participants needs clarification. Did the authors review newborn screening results to ensure that control infants had definitively negative results for VLCADD and other screened conditions? If so, this should be clearly stated in the Methods section to support the validity of group comparisons.

(Answer) As you pointed out, we extracted the results of acylcarnitine analysis from the tandem mass screening test and used these data for group comparison. This information has been added to lines 111-113 of the Methods section.

(Comment 4) Use of Questionnaire and Recall Bias (Line 95):
Given that this is a retrospective study involving caregiver-reported emotional responses, recall bias is a significant concern. The manuscript should address this limitation explicitly, as parental recollection of emotions during the newborn period may be inaccurate or diminished over time.

(Answer) As you pointed out, recall bias is considered a limitation of this study and has been added in lines 236–238 of the Limitations section.

(Comment 5) Emotional Impact Timing and Measurement:
The assessment of emotional responses at a single time point may not fully capture the dynamic nature of psychological reactions. Emotions such as anxiety or regret often fluctuate and may resolve or intensify over time. The authors should acknowledge that evaluating emotional outcomes longitudinally would provide a more comprehensive understanding of psychological burden. This limitation should be discussed.

(Answer) As you pointed out, the previous studies have conducted repeated questionnaire surveys at several-week intervals with caregivers of infants who received false-positive results. In contrast, the present study was not designed as a prospective follow-up study to assess the fluctuations of anxiety over time. This limitation has added in line 240-241.

(Comment 6) Questionnaire Content:
It is unclear whether the questionnaire included items assessing long-term concerns of the parents regarding their child’s growth, development, or health outcomes following the false-positive result. If not, this may represent an important missed opportunity, as ongoing worry is a key component of parental anxiety. Please clarify the scope of the emotional domains evaluated.

(Answer) As you pointed out, the questionnaire did not include items that specifically addressed long-term anxiety. Instead, we used interviews for caregivers who reported experiencing anxiety to explore the nature and causes of their concerns, as well as any temporal changes. During these interviews, we also confirmed—through sources such as maternal and child health handbooks—that none of the infants with false-positive results had subsequently developed serious illnesses or developmental disorders. Furthermore, caregivers’ awareness of their child’s healthy development in everyday contexts helped alleviate their anxiety.

(Comment 7) Emotional Recall Validity:
In assessing emotional reactions such as anxiety or confusion, did the authors consider the possibility that caregivers may have forgotten or minimized their feelings at the time of the event? This issue further underscores the need to address recall bias and emotional adaptation over time.

(Answer) As you pointed out, this study included participants whose children were born over a ten-year period, from 2014 to 2023. Consequently, the interval between the neonatal period and the time of questionnaire completion varied widely, ranging from 9 months to 9 years 6 months. As a result, some caregivers experienced changes in perception over time, while others did not. We consider this variability a limitation of the present study, and this point has been added to lines 238–240 of the Limitations section.

(Comment 8) Breastfeeding Status and VLCADD False Positives:
The rationale for including breastfeeding status in the analysis of VLCADD false positives should be clarified. Is there evidence suggesting that breastfeeding may influence acylcarnitine profiles or screening accuracy? If so, please provide supporting references and explain how this variable was incorporated into the analysis.

(Answer) Thank you for highlighting this important point. As you noted, previous studies have reported the association between breastfeeding and early postnatal weight loss, as well as that between weight loss and false-positive VLCADD results. However, to the best of our knowledge, no studies have directly examined breastfeeding as a contributing factor to elevated acylcarnitine levels. In the present study, we investigated all three aspects simultaneously to explore the potential interrelationships among them.

References: Futatani, T.; Shimao, A.; Ina, S.; Higashiyama, H.; Fujita, S.; Ueno, K.; Igarashi, N.; Hatasaki, K. Capillary Blood Ketone Levels as an Indicator of Inadequate Breast Milk Intake in the Early Neonatal Period. J Pediatr 2017, 191, 76-81

Flaherman, V.J.; Narayan, N.R.; Hartigan-O'Connor, D.; Cabana, M.D.; McCulloch, C.E.; Paul, I.M. The Effect of Early Limited Formula on Breastfeeding, Readmission, and Intestinal Microbiota: A Randomized Clinical Trial. J Pediatr 2018, 196, 84-90 e81

(Comment 9) Table 1 – Descriptive Statistics:
Given the relatively small sample sizes, it would be helpful to report both mean ± SD and median and IQR for continuous variables, especially those with skewed distributions. This will provide a more complete picture of central tendency and variability.

(Answer) As advised, we have reported the data using both median with interquartile range (IQR) and mean with standard deviation.

(Comment 10) Table 1 – Standard Deviation Format:
The SD should be reported using the conventional “±” format. For example, the value “3061.3 (+0.052)” is unclear: does “+0.052” represent the SD, or a deviation from a reference value? If this is the SD, please revise it to “3061.3 ± 0.052.” If it refers to a difference from a normal/reference value, this should be clarified in the table or accompanying legend to avoid misinterpretation.

(Answer) As you suggested, the notation has been revised to “±” as appropriate.

(Comment 11) Table 3 – Formatting and Interpretation:

  • The label “Table 3” should be bold.
  • Additionally, the discussion and interpretation of the questionnaire results (questions  presented in Table 3 are insufficient.

(Answer) As you pointed out, the results corresponding to the questionnaire items in Table 3 were not sufficiently described. We have now been added the results to lines 172-174 and 178–183.

(Comment 12) This study is likely central to understanding the parental emotional experience, yet they are underexplored in the results and discussion. The authors should expand on the meaning, implications, and possible psychological interpretations of these items in the context of false-positive screening results.

(Answer) We apologize for the lack of clarity regarding the objective of this study. This research aimed to examine caregiver anxiety associated with a transient fasting state presumed to be from nutritional insufficiency, and to determine whether this experience influenced their perspectives on feeding practices. The study objective has been revised to reflect this focus more clearly.

(Comment 13) Reference Formatting:
Please ensure that all references are formatted in accordance with the journal’s guidelines.

(Answer) We have confirmed that the study is in compliance with the journal’s submission guidelines.

Reviewer 3 Report

Comments and Suggestions for Authors The topic is important and should be investigated further. Despite the small number of cases, it is important to publish the results so that this potential source of error can be pointed out when communicating a positive screening result. However, the authors should emphasize that the results may be random due to the small number of cases and need to be verified in a larger study population, e.g. in a multicenter study. This applies particularly regarding the correlation between neonatal weight loss and false positive suspicion of VLCADD.

Author Response

(Comment) The topic is important and should be investigated further. Despite the small number of cases, it is important to publish the results so that this potential source of error can be pointed out when communicating a positive screening result. However, the authors should emphasize that the results may be random due to the small number of cases and need to be verified in a larger study population, e.g. in a multicenter study. This applies particularly regarding the correlation between neonatal weight loss and false positive suspicion of VLCADD.

(Answer) Thank you for your comment. As you pointed out, the small sample size limited the generalizability of the findings, and the results should be considered preliminary. Accordingly, we have noted in the Limitations section the need for future studies to re-evaluate the association between neonatal weight loss and false-positive VLCADD results through multicenter case accumulation.

Round 2

Reviewer 1 Report

Comments and Suggestions for Authors

I appreciate the efforts of the authors in responding to the issues I raised in the first review.

This paper is making conclusions on 11 patients with 6 being interviewed. The coverage is small. I acknowledge the rarity of the condition.

Removing parental anxiety in the title is well- received since there were no formal metrics used and the authors admitted they could not use it because of the retrospective nature of the study (2014-2024). Recall bias was an issue.

Authors added statements to clarity further the issues raised in the first review. Also, the limitations are now properly stated in the paper. It is not wise to make conclusions using a small number of patients.

I have minor comments.

I still have a problem with the title.  The title should give a prospective reader a preview of what to expect in the paper.   

I  suggest:

1) removing ‘Characteristics of’ . I do not think the paper provides characteristics… I think the title will hold without this words.

2) limiting the results to Oita Prefecture

If the paper is limited to the Oita Prefecture, it tells the reader that you are sharing your experience in your Prefecture.  Readers will still be interested. There will be no false claims.

Here is my proposed title: Caregivers’ emotional responses triggered by a false-positive VLCADD in newborn screening in Oita Prefecture

Line 26 Abstract: I recommend to just go straight to caregivers’ responses

Line 111: [15] is not needed here

If the authors are keen to pursue this topic, they can consider doing the study prospectively so that answers can be more objective and not based on distant recall. This time, they can use the parental stress index that was mentioned in the paper. It could be one recommendation of the paper.

Comments on the Quality of English Language

Can we have one final review on the English?

Author Response

Comment 1: Removal of “Characteristics of”
Repones 1: As you suggested, we have removed “Characteristics of” from the title to make it more concise and accurate.

Comment 2: Limitation to Oita Prefecture
Repones 2: We have revised the title to indicate that the results are limited to Oita Prefecture, thereby clarifying the scope of our study and avoiding any misleading.

Accordingly, the revised title now reads:
“Caregivers’ emotional responses triggered by a false-positive VLCADD in newborn screening in Oita Prefecture.”

Comment 3:

  1. Revised the abstract (Line 26) to begin directly with caregivers’ responses.
  2. Removed the unnecessary citation [15] at Line 111.

Repones 3: We have corrected them as following your comments:

Comment 4: If the authors are keen to pursue this topic, they can consider doing the study prospectively so that answers can be more objective and not based on distant recall. This time, they can use the parental stress index that was mentioned in the paper. It could be one recommendation of the paper.

Repones 4: We agree with your recommendation for a future prospective study using validated tools such as the parental stress index. We have now emphasized this in the last session of discussion as follows:

Validated instruments such as the Parental Stress Index may serve as appropriate tools to generate reliable evidence to inform future policy and practice.

Thank you once again for your valuable comments.

Reviewer 2 Report

Comments and Suggestions for Authors

Thank you for the thorough revision and for addressing the previous comments appropriately. I have no further suggestions, except to ensure that the formatting strictly follows the journal’s guidelines. Best of luck with your submission.

Author Response

We sincerely thank you for your positive evaluation of our revised manuscript and for your kind words. We will carefully review the formatting once again to ensure that it strictly follows the journal’s guidelines.

We truly appreciate your comments throughout the review process, which has been invaluable in improving our work.